# NiCo_2_S_4_/MoS_2_ Nanocomposites for Long-Life High-Performance Hybrid Supercapacitors

**DOI:** 10.3390/nano13040689

**Published:** 2023-02-10

**Authors:** Le Nhu Minh Tue, Sumanta Sahoo, Ganesh Dhakal, Van Hoa Nguyen, Jintae Lee, Yong Rok Lee, Jae-Jin Shim

**Affiliations:** 1School of Chemical Engineering, Yeungnam University, 280 Daehak-Ro, Gyeongsan 38541, Gyeongbuk, Republic of Korea; 2Department of Chemistry, Nha Trang University, 2 Nguyen Dinh Chieu, Nha Trang, Vietnam

**Keywords:** supercapacitor, NiCo_2_S_4_, MoS_2_, coulombic efficiency, cycling stability

## Abstract

Metal sulfides (MS) and mixed metal sulfides (MTMS) have been considered potential candidates over their metal oxide/mixed metal oxide counterparts in recent years. Herein, one MTMS, i.e., NiCo_2_S_4_, was combined with 2D MS MoS_2_ through a single-step solvothermal process with different morphologies (sheet-like and rod-like) for supercapacitor applications. The resulting electrode exhibited excellent coulombic efficiency, high specific capacitance, superior energy density, and, most importantly, ultra-high cycling stability. In particular, the electrode delivered a capacitance of 2594 F g^−1^ at 0.8 A g^−1^ after 45,000 charge/discharge cycles with a remarkable stability of 192%. Moreover, the corresponding hybrid supercapacitor device displayed an impressive coulombic efficiency of 123% after 20,000 cycles and 118% after 45,000 cycles. In addition, the device also exhibited a decent energy density of 31.9 Wh kg^−1^ and good cycling stability of 102% over 15,000 cycles.

## 1. Introduction

Fossil fuel resources are predicted to be depleted in the next few decades. Therefore, developing renewable and clean energy conversion/storage systems will play a key role in fulfilling the future energy demand. Researchers have been exploring innovative synthetic approaches to fabricate advanced energy storage devices for the last few decades. However, it is challenging to explore easy and inexpensive synthesis methods which are industrially viable. Apart from this, economic efficiency is also one of the most critical factors for long-term operation. In this aspect, supercapacitors are considered favorable, clean electrochemical energy storage devices because of their high power density (*P_D_*), high cycle life, and fast charge/discharge ability [1,2,3,4,5].

Supercapacitors can be ordinarily classified into three groups on the basis of their electrochemical storage kinetics: electrical double-layer capacitor (EDLC), pseudocapacitor, and hybrid capacitor (a combination of EDLC and pseudocapacitor). Typically, pseudocapacitor electrodes are fabricated from transition metal oxides and conducting polymers. On the other hand, EDLCs mostly use carbonaceous materials. In general, EDLC exhibits comparatively lower specific capacitance (*C*_sp_) and energy density (*E_D_*) than pseudocapacitors because of the fast, reversible redox reactions at the electrode-electrolyte interface of electrodes. However, EDLC has good intrinsic conductivity, chemical stability, *P_D_*, and cycle life. Overall, hybrid capacitors are the best choice for next-generation electronic devices, combining the advantages of both EDLC and pseudocapacitors to produce advanced high-performance supercapacitors [6].

Nevertheless, the *E_D_* of commercialized supercapacitors is still lower than that of batteries and fuel cells, despite being higher than that of conventional dielectric capacitors [7,8]. The *E_D_* can be enhanced by altering several strategies such as improving the voltage window of the electrolyte, combining the carbon materials with metal oxide/mixed metal oxides, or integrating different pseudocapacitive/battery-type electrodes [9]. Among the transition metals, Ni and Co are rich in valence electrons. Therefore, such metals considerably impact supercapacitors with their rich electrochemical activity. Moreover, their corresponding MTMS, such as NiCo_2_S_4_, exhibit superior faradaic behavior to those of mono-metallic sulfides counterparts, such as NiS and CoS [10,11,12], and higher electrical conductivity than bimetallic NiCo_2_O_4_ [13]. In addition, the 2D layered compound of Mo, such as MoS_2_, has both the metallic 1T phase and the semiconducting 2H phase [14,15]. Such 2D material also has a small bandgap of 1.2~1.9 eV that can be used in nanoelectronics [16]. It is important to note that the characteristics of MoS_2_ are similar to another carbonaceous 2D material, i.e., graphene. However, MoS_2_ has better capacitive characteristics than graphene. It also displays faster intrinsic ionic conductivity than oxides [17,18].

In the present work, NiCo_2_S_4_ is combined with MoS_2_ to develop a hybrid supercapacitor device. Herein, MoS_2_ is chosen as one of the components as the bulk MoS_2_ is composed of metallic Mo-layers sandwiched between two sulfur layers held together by weak van der Waals interactions. Hence, the electrolyte ions can diffuse quickly and intensely into the material and increase the electrochemical charge storage properties. Interestingly, NiCo_2_S_4_ only uses the anions of the electrolyte in the charging process, whereas MoS_2_ uses the cations of the electrolyte during the discharge process. Later, such charge−discharge characteristics tend to enhance the overall electrochemical properties of the supercapacitors. Benefiting from the MS and MTMS, the electrode exhibited excellent *C*_sp_ after 45,000 charge/discharge cycles. Moreover, the corresponding hybrid device demonstrated good cycling stability over 15,000 cycles and decent *E_D_*.

## 2. Experimental Section

### Synthesis of NiCo_2_S_4_-MoS_2_ Composites

The NiCo_2_S_4_-MoS_2_ composite was synthesized by the one-step solvothermal approach. The precursors, 0.3 g Ni(NO_3_)_2_, 0.6 g Co(NO_3_)_2_, 2 mL C_2_H_8_N_2_ (ethylenediamine), 0.8 g C_2_H_5_NS (thioacetamide), and different amounts (0.05 and 0.15 g) of Na_2_MoO_4_ were dissolved and mixed in 30 mL C_2_H_6_O_2_ (ethylene glycol) with constant stirring and shifted to a Teflon-lined autoclave. The solution-filled autoclave was heated at 200 °C for 15 h and then cooled down. After the completion of the reaction, the material was filtered, washed, and dried at 60 °C for 12 h. Two composites with dissimilar morphologies called NCMS-L (NiCo_2_S_4_/MoS_2_ with low Mo content) (sheet-like) and NCMS-H (NiCo_2_S_4_/MoS_2_ high with Mo content) (rod-like) were produced using low (0.05 g) and high (0.15 g) amounts of Na_2_MoO_4_, respectively.

Furthermore, bare NiCo_2_S_4_ and bare MoS_2_ were also synthesized using the above-mentioned procedures in the absence of Mo^6+^ for bare NiCo_2_S_4_ and Ni^2+^ and Co^2+^ for bare MoS_2_, respectively.

## 3. Results and Discussion

### 3.1. Synthesis Mechanism of NiCo_2_S_4_/MoS_2_ Composite

The current work is focused on developing two different morphologies of the composite of NiCo_2_S_4_ and MoS_2_. A schematic diagram demonstrates the formation mechanism of these two morphologies (Figure 1). Initially, Ni(NO_3_)_2_, Co(NO_3_)_2_, and Na_2_MoO_4_ were dissolved in ethylene glycol to form Ni^2+^, Co^2+^, and MoO_4_^2−^ ions. It is important to note that the rapid transportation of MoO_4_^2−^ was restricted by its high molar mass and the high viscosity of ethylene glycol solvent. Additionally, the formation of stable NiCo_2_S_4_ took longer time than MoS_2_. Therefore, MoO_4_^2−^ was first combined with H_2_S to form amorphous MoS_2_. Therefore, no apparent peaks of MoS_2_ are observed in the XRD patterns of the composite materials (Figure 2a−c). With the assistance of the –NH_2_ group in thioacetamide, Ni^2+^ and Co^2+^ formed stable complexes on the surface of MoS_2_ and then reacted with H_2_S to produce NiCo_2_S_4_ [19]. On the other hand, the morphology of composite material changed from sheet-like to rod-like by increasing the concentration of the precursor Na_2_MoO_4_.2H_2_O from 0.05 g to 0.15 g (Figure 3a−d). Probable reactions during the solvothermal process are shown below:CH_3_CSNH_2_ + H_2_O → CH_3_CONH_2_ + H_2_S(1)
MoO_4_^2−^ + 3H_2_S →MoS_2_ + 3H_2_ + SO_4_^2−^(2)
Ni^2+^ + 2Co^2+^ + 4H_2_S → NiCo_2_S_4_ + 4H_2_(3)

The change in morphology with altering the concentration of MoO_4_^2−^ can be attributed to the change in the direction of diffusion of MoO_4_^2−^ during its ion-exchange process with S^2−^. The inward diffusion of MoO_4_^2−^ produced sheet-like morphology, whereas the outward diffusion resulted in the synthesis of a rod-like composite [20].

### 3.2. Characterization and Morphology of NiCo_2_S_4_/MoS_2_ Composite

The crystallinity of composite materials was examined by XRD analysis (Figure 2a−c). The XRD pattern of bare NiCo_2_S_4_ demonstrates all the characteristic peaks for the NiCo_2_S_4_ phase (JCPDS card No. 43-1477). On the other hand, for NCMS-L, the XRD peaks are observed at 17.02, 26.82, 31.54, 37.76, 50.23, and 55.12° 2θ, corresponding to the (111), (220), (311), (400), (511), and (440) planes of NiCo_2_S_4_, respectively (Figure 2b). However, the HRTEM analysis reveals the presence of the (002) plane of MoS_2_, according to the JCPDS card No.37-1492 (Appendix A). Moreover, the concentration of precursor Mo^6+^ was low compared to Ni^2+^ and Co^2+^, which generates dominant peaks of NiCo_2_S_4_ to suppress the peaks of MoS_2_. The XRD pattern of bare amorphous MoS_2_ confirms the amorphous nature (Figure 2a). In addition, the absence of any secondary peak in the XRD pattern of the composites indicates their high purity. A similar XRD pattern has also been observed for NCMS-H (Figure 2c). The morphologies of bare NiCo_2_S_4_ and MoS_2_ and their composites were analyzed by FESEM, as shown in Figure 3a−d. Both bare NiCo_2_S_4_ and MoS_2_ display the presence of aggregated clusters. However, compared with MoS_2_, NiCo_2_S_4_ shows bigger clusters. On the other hand, composite materials NCMS-L and NCMS-H exhibit sheet-like and rod-like morphology, respectively. From the morphological analysis, it is confirmed that NCMS-L displays higher porosity than NCMS-H. This fact also explains why the capacitance retention of NCMS-L was better than NCMS-H after cycling tests (Figure 7d and Appendix A) [13,21,22]. Furthermore, TEM analysis was performed to examine the microstructure of the electrode materials. As shown in Figure 4a, NCMS-L displays a porous sheet-like structure. On the contrary, NCMS-H shows a rod-like structure (Figure 4b). Therefore, it is essential to note that the SEM images support the TEM analysis result. On the other hand, Figure 4c,d shows the structure of MoS_2_ and NiCo_2_S_4_, respectively. However, the NiCo_2_S_4_ particles are found to be larger and denser than MoS_2_. Owing to the small-sized particles and amorphous structure of MoS_2_, the electrolyte can pass easily inside the composite material and enhance the redox reaction to deliver high *C*_sp_.

The elemental composition and chemical state in NCMS-L were evaluated by XPS, as presented in Figure 5a−e. The survey spectrum confirms the presence of metal components and the other corresponding elements, such as O and S (Figure 5a). The deconvoluted Ni 2p core-level spectrum shows four peaks. Herein, two spin-orbit doublets correspond to Ni^2+^ and Ni^3+^ [23]. Furthermore, two shake-up satellites (identified as “Sat.”) have also been observed (Figure 5b). On the other hand, two spin-orbit doublets in Co 2p spectrum can be designated as Co^2+^ and Co^3+^ [24]. Moreover, one shake-up satellite has also been observed (Figure 5c). The S 2p spectrum is deconvoluted into two significant peaks and a shake-up satellite peak with binding energies of 162.8 and 161.6 eV, corresponding to S 2p_1/2_ and S 2p_3/2_, respectively, as shown in Figure 5e [25].

The Mo 3d XP spectrum in Figure 5d shows two strong peaks at 228.8 and 232.4 eV, which can be assigned to the Mo 3d_5/2_ and Mo 3d_3/2_ doublet, respectively. The peak at 226.2 eV is assigned to S 2s [20]. Furthermore, to determine the presence of Mo in the composite material, EDX elemental mapping was performed for NCMS-L through HR-TEM analysis (Figure 6 and Figure S2). As expected, the concentration of Mo is lower than the other elements, Ni, Co, and S. However, all elements are distributed uniformly. From EDX analysis, it can be concluded that MoS_2_ can cooperate and enhance the electrochemical properties of NiCo_2_S_4_ to obtain promising electrochemical performance.

### 3.3. Electrochemical Properties of NiCo_2_S_4_/MoS_2_ Nanocomposite

The CV (cyclic voltammetry), GCD (galvanostatic charge/discharge), and EIS (electrochemical Impedance Spectroscopy) tests were performed to explore the electrochemical properties of NCMS-L, NCMS-H, bare MoS_2_, and bare NiCo_2_S_4_ electrodes in a 3 M KOH electrolyte. Figure 7a represents the CV curves of NCMS-L within the potential limit of 0 to 0.6 V at various scan rates of 5–100 mV s^–1^. The CV curves reveal faradaic behavior, in which a pair of prominent redox peaks are observed. Similarly, NiCo_2_S_4_ and MoS_2_ electrodes also have a pair of redox peaks at different scan rates (Appendix A). The corresponding redox reactions can be expressed as follows [14,15,21,26,27,28,29,30]:CoS + OH^−^↔ CoSOH + e^−^(4)
CoSOH + OH^−^ ↔ CoSO + H_2_O + e^−^(5)
NiS + OH^−^ ↔ NiSOH + e^−^(6)
MoS_2_ + K^+^ + e^−^ ↔ MoS-SK(7)
MoS_2_ + K^+^ + e^−^ ↔ MoS_2_-K(8)

The GCD test of NCMS-L, bare NiCo_2_S_4_, and bare MoS_2_ are performed at the constant currents of 1, 3, 5, 8, and 10 mA (or corresponding current densities of 0.8, 2.3, 3.8, 6.2, and 7.7 A g^−1^). Based on the GCD and CV curves, the *C*_sp_ of the electrodes was calculated using the following equations:(9)$$C_{s}=\frac{It}{m∆V}$$
(10)$$C_{s}=\frac{\int i\left(V\right)dV}{2vm∆V}$$
where *I* is the response constant current (A); *t* is the discharge time (s); *ΔV* is the potential window of the CV or GCD curves (V); *m* is the mass of active material (g); *i* and *V* are current and potential in the CV curve (A and V), respectively, and *ν* is the scan rate (V s^–1^) [31].

The *C*_sp_ of NCMS-L was calculated to 1390, 1000, 864, 756, and 698 F g^−1^ at the currents of 1, 3, 5, 8, and 10 mA (corresponding current densities of 0.8, 2.3, 3.8, 6.2, and 7.7 A g^−1^), respectively (Figure 7b). It is important to note that the GCD profiles indicate significant deviation from the ideal rectangular shapes, indicating the Faradaic characteristics. The cycling stability of the electrode was assessed by continuing the CV test for 45,000 cycles at the scan rate of 100 mV s^−1^, as presented in Figure 7c,d. The electrode exhibited superior capacitance retention of 135% over the first 8000 cycles. However, after these initial cycles, the capacitance retention gradually increased to 149% for up to 35,000 cycles, which is associated with the activation of the electrode material. For further activation of the electrode and to check the electrode’s feasibility for long-term stability, the cycling process was stopped for one week with the electrode soaked in the electrolyte solution. The cycling test continued up to the 45,000th cycle. We found that soaking the electrodes in the electrolyte for a long time (after activation) was quite favorable for enhanced cycling stability. Such drastic enhancement in electrochemical performance is associated with the electrochemical reconstruction of the electrode materials during cycling and soaking the electrode in electrolyte solution for a long time [32]. It is important to note that, after the soaking process, the capacitance retention increased to 207% at the 38,000th cycle, which can be associated with the re-activation of the electrode. However, the capacitance then gradually decreased and reached 192% of its initial capacitance at the 45,000th cycle due to the structural breakdown, as depicted in Appendix A.

The cycling stability increased to 207% at the 38,000th cycle; after that, it decreased slightly to 192% at the 45,000th cycle. The GCD test examined the retention of the specific capacitance of NCMS-L at 1 mA (Figure 7e). The *C*_sp_ was enhanced from 1390 to 2293 F g^−1^ at the 20,000th cycle and 2594 F g^−1^ at the 45,000th cycle. Moreover, the coulombic efficiency increased from 73% to 123% at the 20,000th cycle and then slightly decreased to 118% at the 45,000th cycle. Figure 4f shows the corresponding Nyquist plots at the 1st, 35,000th, and 45,000th cycles. The intercept at the *y*-axis in the high-frequency region that signifies the equivalent series resistance (ESR) was 0.45 Ω (1st cycle), 0.49 Ω (35,000th cycle), and 0.39 Ω (45,000th cycle), respectively. The increase and decrease in ESR over the cycles are agreed well with the change in capacitance over the long-term cycling stability. Similar trends have also been found for NiCo_2_S_4_ and MoS_2_, as shown in Appendix A.

Comparing CV curves of MoS_2_, NiCo_2_S_4_, and NCMS-L at a scan rate of 100 mV s^−1^ displays the highest current response for the composite (Figure 8a). The comparative GCD plots at the current of 1 mA also designate a higher discharge time of NCMS-L than its monometallic and bimetallic counterparts (Figure 8b). Figure 8c represents the CV curves of NiCo_2_S_4_ at different cycles at the constant scan rate of 100 mV s^−1^. Compared to the 1st cycle, a significant change in CV pattern is observed at the 5000th cycle. This fact indicates the gradual activation of the electrode material. However, the CV curves’ redox characteristics do not alter after 1000 cycles, indicating good electrochemical stability. Most importantly, the electrode displayed a good cycling stability of 147% after 5000 cycles (Figure 8d). A similar pattern has also been observed for the MoS_2_ electrode (Figure 8e). Nevertheless, the MoS_2_ displayed better cycling stability (248%) than NiCo_2_S_4_ over 5000 cycles (Figure 8f). Similar to other supercapacitor electrodes, the area under the CV profiles increases with increasing the scan rate for NiCo_2_S_4_ (Appendix A). The faradaic charge storage characteristics are visible from the GCD profiles at different current densities (Appendix A). It is important to note that the discharge time was increased after 5000 cycles (Appendix A).

Likewise, the capacitance of NiCo_2_S_4_ was increased from 947 to 1121 F g^−1^ after 5000 cycles. Appendix A−d represents the electrochemical data for MoS_2_. Similar to NiCo_2_S_4_, the faradaic charge storage ability of MoS_2_ is visible from their CV curves and GCD profiles (Appendix A. However, the cycling stability of MoS_2_ drastically increased from 323 to 1100 F g^−1^ after 5000 cycles. Overall, MoS_2_, NiCo_2_S_4_, and NCMS-L showed the *C*_sp_s of 323, 947, and 1390 F g^−1^, respectively. The better capacitive performance of NCMS-L can be ascribed to the strong synergistic contribution of its components, i.e., MoS_2_ and NiCo_2_S_4_. For NCMS-L, Ni^2+^ and Co^3+^ are combined with OH^−^ to produce Ni^3+^ and Co^4+^ during charging. However, K^+^ is inserted into the active material and has a redox reaction on the surface of MoS_2_ during discharging [15]. These results explain the increase in *C*_sp_ of NCMS-L and a considerable increase in the coulombic efficiency from 73% to 123% over 20,000 cycles. On the other hand, it takes a long time for K^+^ ions to penetrate the structure of NCMS-L and have a redox reaction with MoS_2_. After 35,000 cycles, the cycling process was suspended for one week to ensure the complete diffusion of the electrolyte ions into the active material, obtaining enhanced capacitance retention of 192%.

To understand the individual contributions of NiCo_2_S_4_ and MoS_2_ on the cycling stability of NCMS-L, an EIS analysis was performed. After 5000 cycles, the ESR of bare NiCo_2_S_4_ increased from 0.53 to 0.6 Ω (Appendix A), whereas it decreased from 1.86 to 0.58 Ω for bare MoS_2_ (Appendix A). The significant decrease in ESR for MoS_2_ is well supported by the substantial enhancement of discharge time, as shown in Appendix A. The morphological analysis of these electrodes after the cycle test reveals a significant change in the structure of MoS_2_ from clusters to coral (Appendix A). However, no significant change in morphology has been observed for NiCo_2_S_4_ (Appendix A).

Interestingly, during the pause of the cycling test for two weeks, the morphology of MoS_2_ was further changed from coral to chunks (Appendix A). However, no significant change in the morphology of NiCo_2_S_4_ has been observed after the pause of the cycle test (Appendix A). This phenomenon indicates that MoS_2_ undergoes drastic structural reformation during the cycling test as well as during the pause of the cycle test. This fact also supports the electrochemical reconstruction of NCMS-L during the cycle test. Therefore, it can be concluded that MoS_2_ has majorly contributed to the enhanced electrochemical performance of NCMS-L over long-term cycle life. The enhanced cycling stability of MoS_2_ is the driving force of such boosted cycling stability of the composite.

To further estimate the prospective of the NCMS-L electrode for practical applications, an asymmetric supercapacitor device was designed with activated carbon (AC) as the negative electrode, NCMS-L as the positive electrode, and a 3 M KOH aqueous solution as the electrolyte.

The mass of the negative and positive electrodes was adjusted based on the charge balance theory (q^+^ = q^−^), as shown in the following equation:(11)$$q=I∆t=C_{s}m∆V$$

Therefore, the mass ratio of the electrode materials was calculated using Equation (12).
(12)$$\;\frac{m^{+}}{m^{-}}=\frac{C_{s}{}^{-}∆V^{-}}{C_{s}{}^{+}∆V^{+}}$$
where $C_{s}$, ∆V, and m are the *C*_sp_, potential window, and mass of active materials, respectively. The mass ratio of NCMS-L and AC was 0.26, and the total mass of the two electrodes was 6.3 mg.

Figure 9a represents the CV curves of the NCMS-L//AC hybrid supercapacitor device within potential limit of 0 to 1.6 V at different scan rates from 5 to 100 mV s^−1^. The CV curves indicate the presence of redox peaks, indicating the Faradaic charge storage mechanism. Figure 9b presents the GCD profiles at various current densities of 0.16, 0.32, 0.47, 0.79, 1.27, and 1.59 A g^−1^, respectively. The maximum *C*_sp_ of the device was calculated to be 90 F g^−1^ at a CD of 0.16 A g^−1^. The device also delivered a high-rate capability of 64% after a 10-fold increase in the CD (Figure 9c). As shown in Figure 9d, the device exhibited a long cycle life over 15,000 cycles, which is better than some of the reported devices (Appendix A). The device retained 111% of its initial capacitance during the first 1500 cycles. The initial increase in capacitance can be attributed to the steady activation of electrode material. The device exhibited a capacitance retention of 102% after 15,000 cycles. Moreover, it also displayed 98% retention of its initial coulombic efficiency after 15,000 cycles. The enhanced cycling stability of the device can be ascribed to the porous nature of the positive electrode, the synergistic contribution of the individual components of the composite, and the better chemical stability of the carbon black (negative electrode). The device also displayed higher *E_D_* than the other related devices, as shown in the Ragone plot (Figure 9e) [13,33,34,35]. Based on the equations, *E_D_* = 0.5 CV^2^ and *P_D_* = *E_D_*/*t*, the maximum *E_D_* was calculated to be 31.9 W h kg^−1^ at a *P_D_* of 0.13 kW kg^−1^, and the highest *P_D_* was 1.26 kW kg^−1^ at the corresponding *E_D_* of 20.7 Wh kg^−1^. The ESR of the device in the first cycle was approximately reduced to half that of the three-electrode system with a value of 0.22 Ω. However, it was further increased to 0.32 Ω after 15,000 cycles (Figure 9f).

The electrode displayed decent capacitance, low ESR, and enhanced cycling stability. Such enhanced electrochemical performance of NCMS-L can be attributed to the following factors:The synergistic contribution of each composite component is highly responsible for such enhanced electrochemical performance. The combination of highly capacitive NiCo_2_S_4_ and highly conductive MoS_2_ is feasible for fabricating high-performance supercapacitor electrodes.The enhanced cycling stability of the composite can be attributed to its electrochemical reconstruction during the cycling test.The porous morphology of the composite (NCMS-L) is favorable for the easy transport of electrolyte ions during the electrochemical test.The enhanced mechanical strength of MoS_2_ restricted the structural deformation of the electrode during the long-term cycling test.

## 4. Conclusions

Advanced composite electrodes based on NiCo_2_S_4_ and MoS_2_ with different morphologies have been synthesized through facile solvothermal processes. Benefiting from a rich faradaic charge storage mechanism, the electrodes displayed a promising electrochemical performance in high capacitance and low ESR. Nevertheless, the composite electrode also exhibited long-term cycle life with negligible capacitance fading. Overall, this composite material can be designated one of the striking candidates for future energy storage devices.

## Figures and Tables

**Figure 1 nanomaterials-13-00689-f001:**
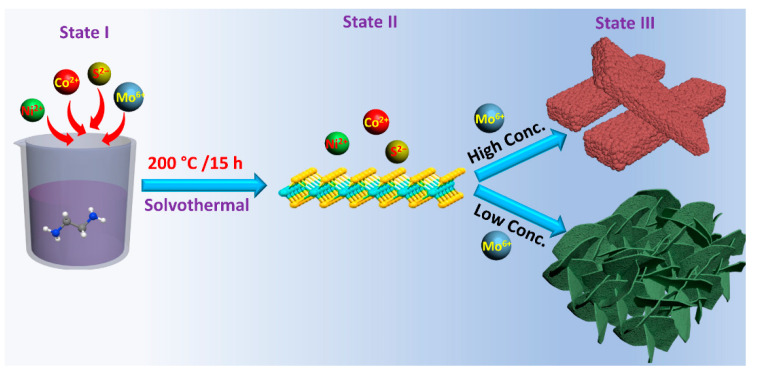
Schematic diagram of the NCMS (State I demonstrates the mixing of all precursors. State II designates the formation of MoS_2_. State III demonstrates the formation of NiCo_2_S_4_ over MoS_2_ with sheet- and rod-like morphologies).

**Figure 2 nanomaterials-13-00689-f002:**
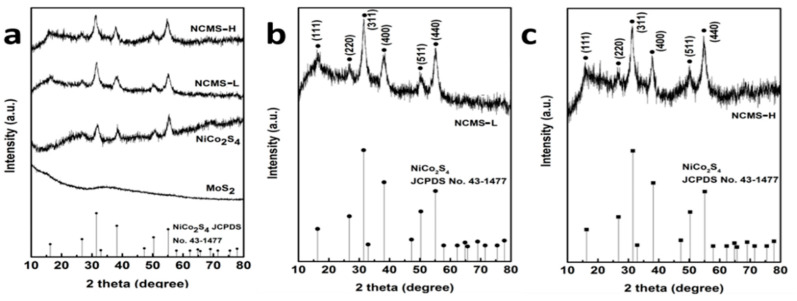
XRD patterns of the composite material NCMS-L, NCMS-H, NiCo_2_S_4_, and MoS_2_ (**a**); XRD patterns of NCMS-L (**b**) and NCMS-H (**c**).

**Figure 3 nanomaterials-13-00689-f003:**
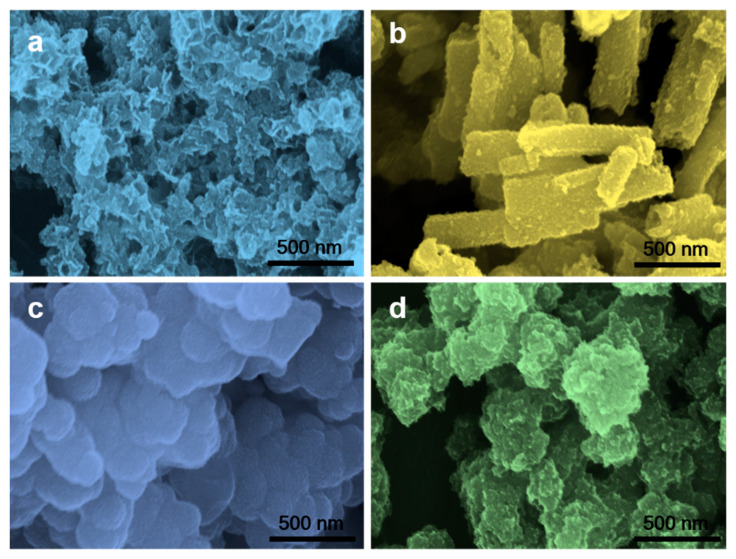
High-magnification SEM images of (**a**) NCMS-L, (**b**) NCMS-H, (**c**) MoS_2_, and (**d**) NiCo_2_S_4_.

**Figure 4 nanomaterials-13-00689-f004:**
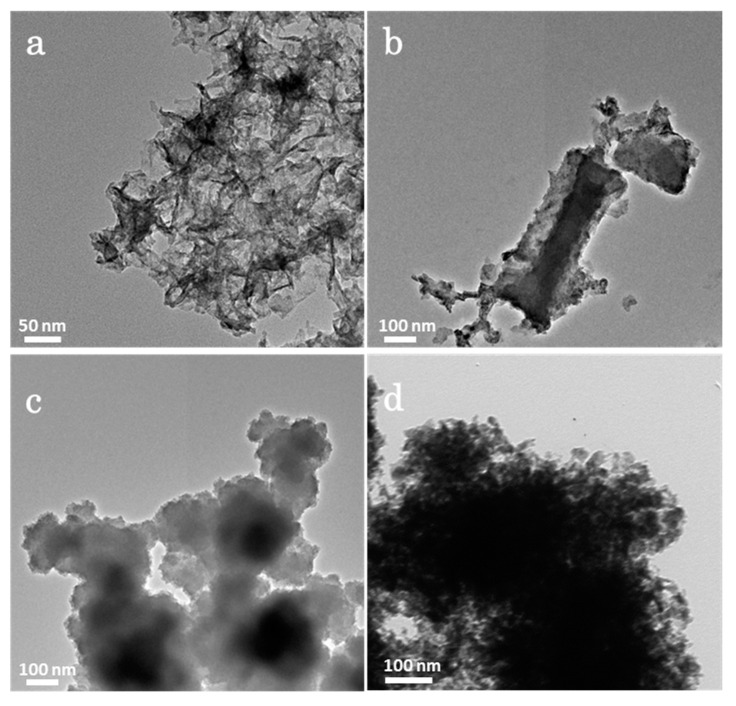
TEM images of (**a**) NCMS-L, (**b**) NCMS-H, (**c**) MoS_2_, and (**d**) NiCo_2_S_4_.

**Figure 5 nanomaterials-13-00689-f005:**
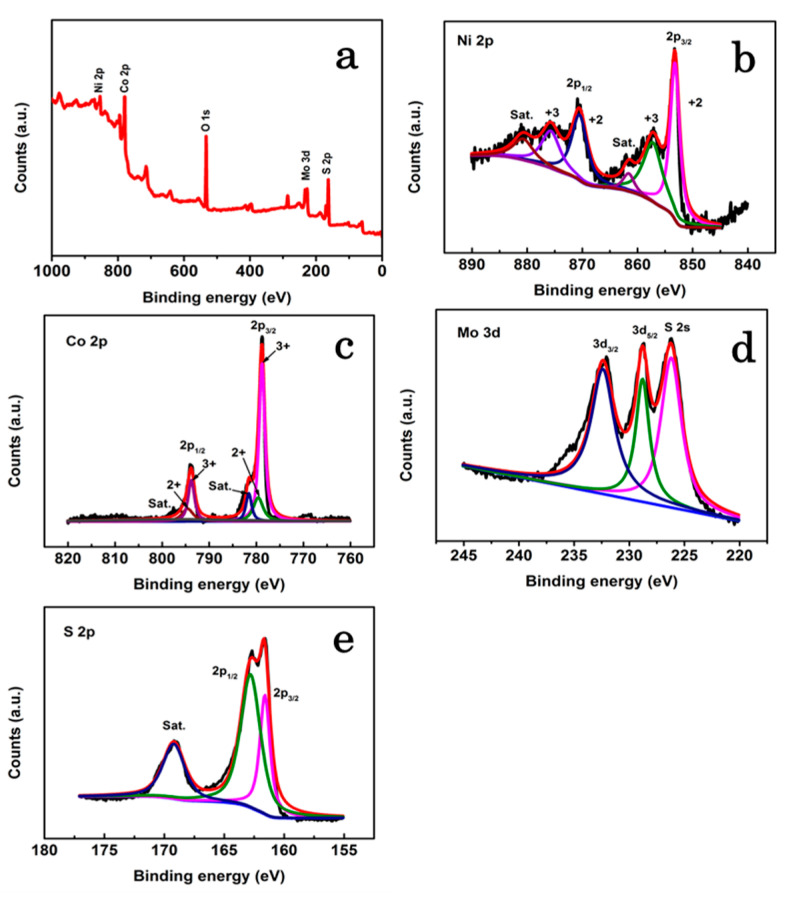
XPS analysis of NCMS-L: (**a**) Survey XPS pattern, high-resolution XPS spectra of (**b**) Ni 2p, (**c**) Co 2p, (**d**) Mo 3d, and (**e**) S 2p.

**Figure 6 nanomaterials-13-00689-f006:**
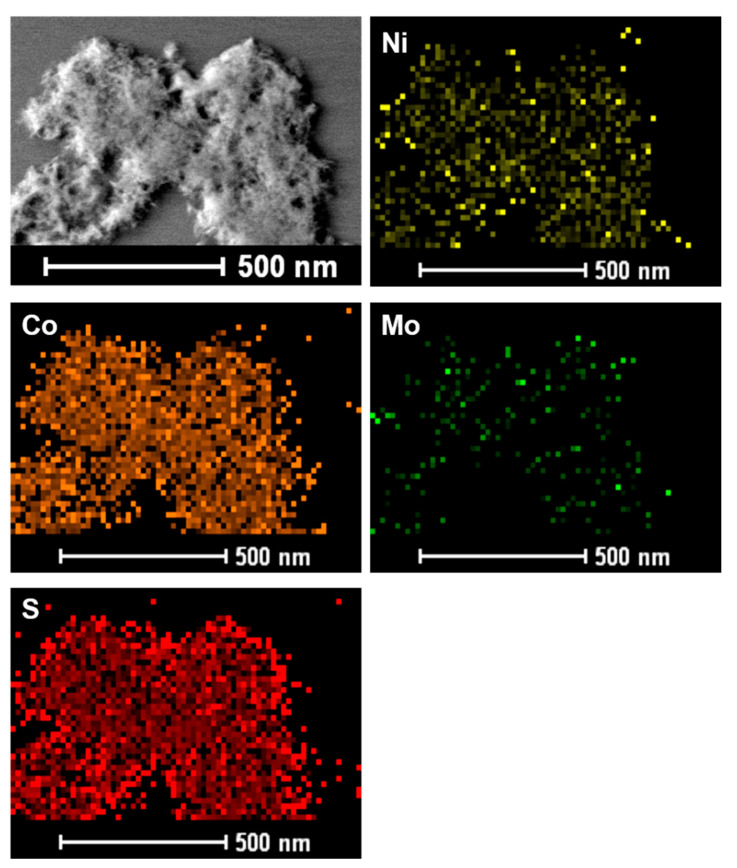
EDX element mapping images of NCMS-L indicating the elemental distribution of Ni, Co, Mo, and S.

**Figure 7 nanomaterials-13-00689-f007:**
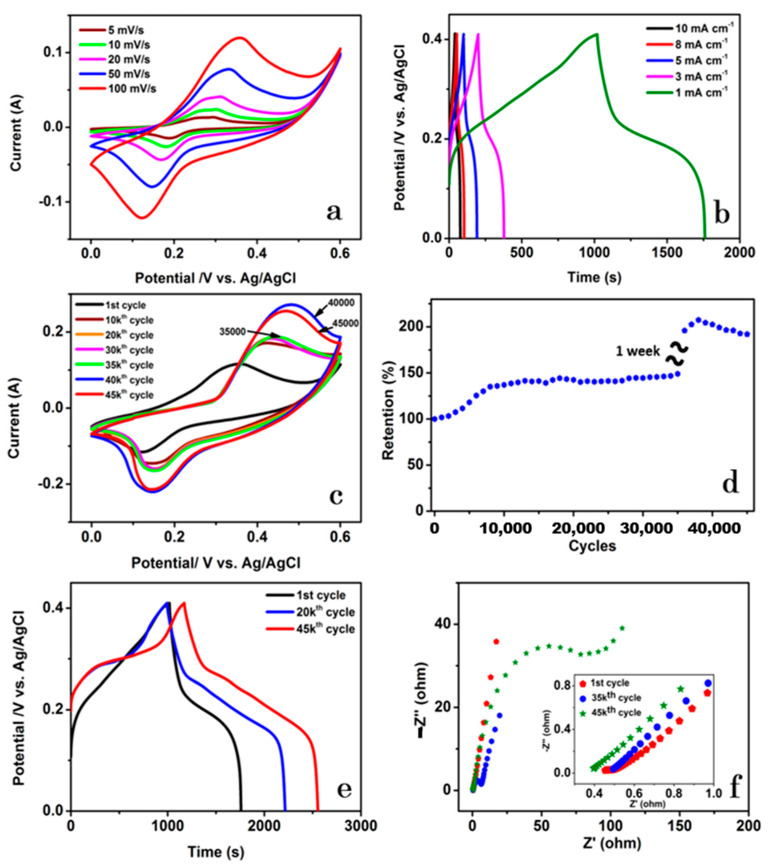
Electrochemical properties of NCMS-L: (**a**) CV curves at different scan rates, (**b**) GCD profiles at currents ranging from 1 mA to 10 mA with a current density (CD) of 0.8, 2.3, 3.8, 6.2, and 7.7 A g^−1^, respectively, (**c**) CV curves at various cycles with a scan rate of 100 mV s^−1^, (**d**) Retention during 45,000 cycles by CV at the scan rate of 100 mV s^−1^, (**e**) GCD profiles at various cycles, and (**f**) Nyquist plots at 1st, 35,000th, and 45,000th cycle.

**Figure 8 nanomaterials-13-00689-f008:**
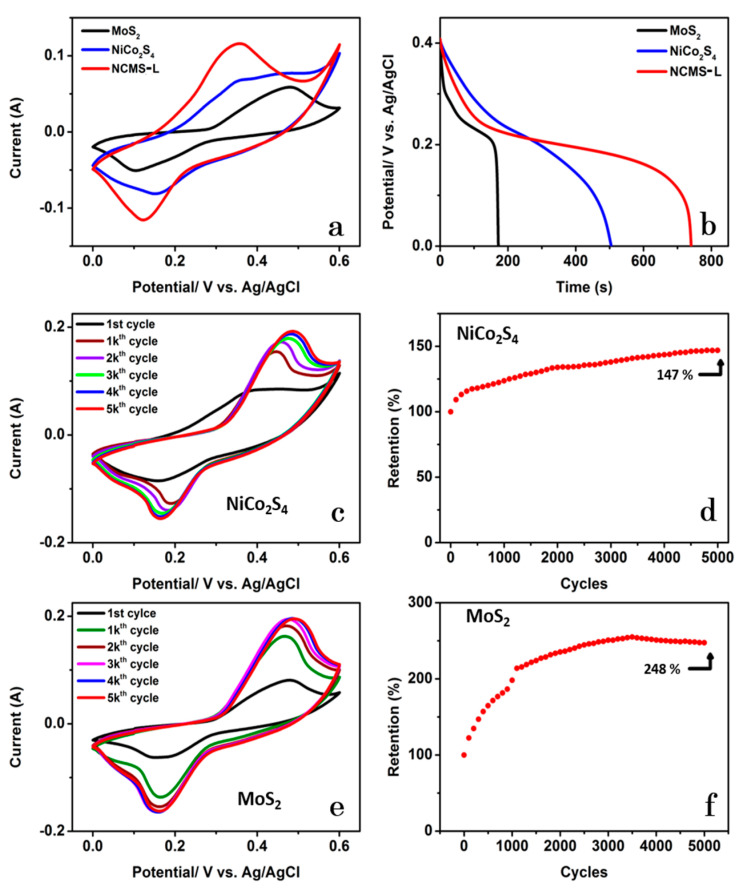
(**a**) Comparative CV curves of MoS_2_, NiCo_2_S_4_, and NCMS-L at a scan rate of 100 mV s^−1^, (**b**) GCD curves of MoS_2_, NiCo_2_S_4_, and NCMS-L at 1 mA (CD of 0.8 A g^−1^); CV curves of (**c**) NiCo_2_S_4_ and (**e**) MoS_2_ at various cycles at a scan rate of 100 mV s^−1^; (**d**,**f**) Retention of NiCo_2_S_4_ and MoS_2_ after 5000 cycles by CV with a scan rate of 100 mV s^−1^.

**Figure 9 nanomaterials-13-00689-f009:**
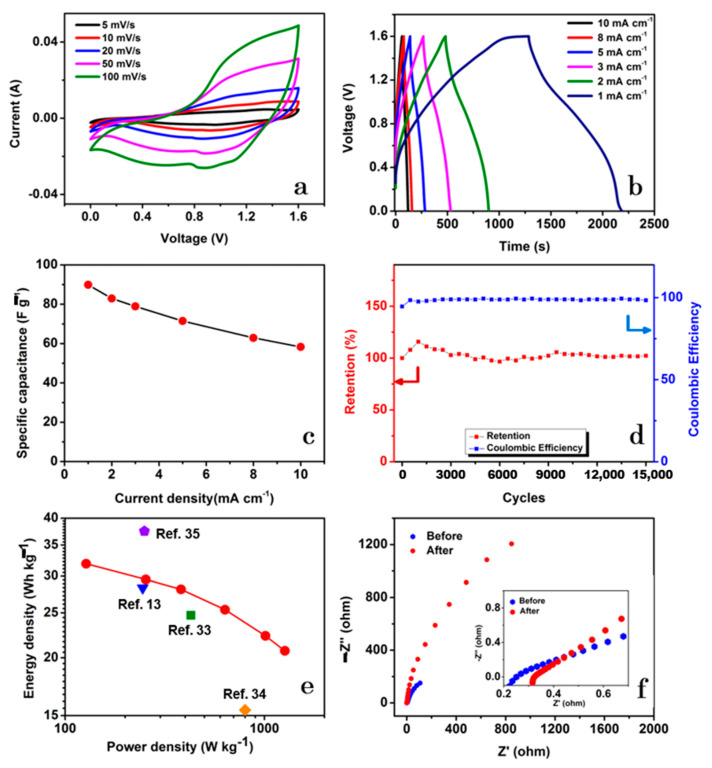
Electrochemical performance of the hybrid supercapacitor device of NCMS-L and activated carbon: (**a**) CV profiles at various scan rates, (**b**) GCD curves at different currents from 1 mA to 10 mA with a CD of 0.16, 0.32, 0.47, 0.79, 1.27, and 1.59 A g^−1^, respectively, (**c**) variation of *C*_sp_ with the change in current densities, (**d**) capacitance retention during 15,000 cycles at the CD of 6.35 A g^−1^, (**e**) the Ragone plot, and (**f**) Nyquist plots at the frequency range of 0.01 Hz to 100 kHz before and after 15,000 cycles.

## Data Availability

The data presented in this study are available in the lab research notebook at Yeungnam University.

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
