# Peer review of "NiCo2S4/MoS2 Nanocomposites for Long-Life High-Performance Hybrid Supercapacitors"

_nanomaterials, 2023, doi:10.3390/nano13040689_

Round 1

Reviewer 1 Report

The manuscript reported the preparation of NiCo2S4/MoS2 Nanocomposites for Long-life High-performance Hybrid Supercapacitors. I recommend this manuscript for publication after addressing serious concers as as listed below .

1.     It is very hard to believe that the electrode delivered a capacitance of 2,594 F g−1 at 0.8 A g−1 after 45,000 charge/discharge cycles with remarkable stability of 192%. Moreover, there is no sufficient explanation as to how the electrode material can achieve such a high specific capacity and durability.

2.     As authors claim MTMS, i.e., NiCo2S4, was combined with the 2D MS MoS2 through a single-step solvothermal process with different morphologies (sheet-like and rod-like) for supercapacitor applications, what is the chemistry behind such different morphologies? How does change in concentration affect the morphology, must be explained properly.

3.     Capacitive contribution must be included.

4.     Authors claim that the porous morphology of the composite (NCMS-L) is favorable for the easy transport of electrolyte ions during the electrochemical test though there is no BET analysis and even SEM and TEM analysis as shown in figures 3 and 4 are not in agreement with authors’ claim.

5.     More importantly, the manuscript did not be well arranged, and had a large numbers of grammar errors and unsuitable description

Author Response

Reviewer #1

Comment 1: It is very hard to believe that the electrode delivered a capacitance of 2,594 F g−1 at 0.8 A g−1 after 45,000 charge/discharge cycles with remarkable stability of 192%. Moreover, there is no sufficient explanation as to how the electrode material can achieve such a high specific capacity and durability.

Response: The enhanced cycling stability of the composite was because of the electrochemical reconstruction during cycling. The relevant reference has been added to the manuscript (Ref. No. 32). More explanations have also been included in the revised manuscript. We hope the reviewer will be satisfied with the explanation.  

Comment 2: As authors claim MTMS, i.e., NiCo2S4, was combined with the 2D MS MoS2 through a single-step solvothermal process with different morphologies (sheet-like and rod-like) for supercapacitor applications, what is the chemistry behind such different morphologies? How does change in concentration affect the morphology, must be explained properly.

Response: Thanks for the valuable comment. The explanation of morphological change has been added to the revised manuscript with a related reference (p. 3).  

Comment 3: Capacitive contribution must be included.

Response: I agree with you that discussing the capacitive contribution is a trend nowadays and also helpful. However, please understand that this study was done 6 years ago. The student who had worked this already returned to his home country a long time back. Recently, his draft and graphs have been modified by a research professor. Therefore, it is very difficult for me to get the original data from the former student and ask the research professor to do this kind of calculation after reading the data from the graph.

Comment 4: Authors claim that the porous morphology of the composite (NCMS-L) is favorable for the easy transport of electrolyte ions during the electrochemical test though there is no BET analysis and even SEM and TEM analysis as shown in figures 3 and 4 are not in agreement with authors’ claim.

Response: Figures 3 and 4 show that the SEM and TEM images of NCMS-L carry sheet-like morphology with high porosity. Many previous reports have confirmed that such porous sheet-like morphology is favorable for the easy transport of electrolyte ions during the electrochemical reactions (Ref. 1–3). As I explained in the previous answer, we cannot get the sample the former student made. Even though we get it, the structure and morphology could have been changed. Therefore, the BET surface area can not be obtained.

1) Lokhande, P.E., Pawar, K. and Chavan, U.S., “Chemically deposited ultrathin α-Ni (OH) 2 nanosheet using surfactant on Ni foam for high-performance supercapacitor application,” Mater. Sci. Energy Technol., 1(2), 166-170 (2018).

2) Wang, R., and Yan, X., “Superior asymmetric supercapacitor based on Ni-Co oxide nanosheets and carbon nanorods,” Sci. Rep., 4(1), 1-9 (2014).

3) Huang, M., Li, F., Ji, J.Y., Zhang, Y.X., Zhao, X.L. and Gao, X., “Facile synthesis of single-crystalline NiO nanosheet arrays on Ni foam for high-performance supercapacitors,” CrystEngComm, 16(14), 2878-2884 (2014).   

Comment 5: More importantly, the manuscript did not be well arranged, and had a large numbers of grammar errors and unsuitable description.

Response: As I have mentioned, the draft of this manuscript was prepared by a student. Though a research professor modified it, many flaws might have remained. Therefore, the manuscript has been significantly modified again. The grammatical errors have been corrected, and the unsuitable descriptions have been removed.

Reviewer 2 Report

In this manuscript, through a one-step solvothermal process, the authors successfully prepared NiCo2S4-MoS2 composites with different morphologies (sheet-like and rod-like), which showed excellent electrochemical performance for supercapacitor applications. Combining a large amount of experimental data, the authors demonstrated that the composite electrode does have a long-term cycle life with nice performance. This study provides promising material and deserves the attention of relevant researchers. Therefore, I recommend the acceptance of this work after the following issues are addressed.

1.       In Figure 2, the characteristic peaks such as peaks corresponding to (711), (642), and (731) planes, which are mentioned to compare the crystallinity between NCMS-L and NCMS-H, are hard to distinguish. The refinement of XRD may be provided if possible.

2.       In the experimental section, detailed test instruments used for electrochemical tests should be given as well as for material characterization. Furthermore, the way to test the electrochemical properties should be depicted in detail.

3.       After being stopped for one week during the cycling process, as shown in Figure 7d, the retention of NCMS-L increased initially while seems to have a tendency to continue declining after 45000 cycles. Therefore, the claim that the capacitance of NSMS-L can hold to 192% after 45000 cycles without prerequisite description is challenging. A longer-term cycle of data should be given to support this opinion, or the pause can be conducted earlier.

4.       Since the authors set up two products with different Mo contents that generated different morphologies, the electrochemical properties of NCMS-H should be given to compare with NCMS-L in the section results and discussion.

5.       The manuscript should be revised thoroughly because there are several inappropriate details. For example, the ‘NCMS-H’ in line 123, the ‘NoiCo2S4’ in line 331, etc.

6.       The prepared composites have different energy storage mechanisms in the process of charge and discharge. That will have a certain impact on calculation of specific capacitance or coulomb efficiency, please give more details of the above electrochemical data calculation. Specifically, it is not normal for coulombic efficiency to be greater than 100%. Please give a reasonable explanation.

7.       XPS analysis was performed in Fig.5 to obtain the valence state of Ni/Co/Mo/S, please provide evidence or relevant reference.

8.       In line 256 of the text, the authors described the oxidation processes of Ni2+ and Co3+, which were contradictory with the reaction mechanisms of Ni2+ and Co2+ given in the equations (4-6). In addition, XPS analyses show bivalent and trivalent Ni/Co, whether both bivalent and trivalent Ni/Co will react? Is the Faradaic reaction potential of cobalt sulfide consistent with nickel sulfide?

9.       Suggest the authors provide pseudo-capacitance calculation and b-value fitting to analyze the capacitance contribution.

10.     The authors should also provide the information about the mechanism of hybrid asymmetric supercapacitor.

Author Response

Reviewer #2

Comment 1: In Figure 2, the characteristic peaks such as peaks corresponding to (711), (642), and (731) planes, which are mentioned to compare the crystallinity between NCMS-L and NCMS-H, are hard to distinguish. The refinement of XRD may be provided if possible.

Response: I agree that the XRD patterns are unclear, and the characteristic peaks with such low intensity can not distinguish the crystallinity between NCMS-H and NCMS-L. We changed to focus on the major XRD peaks of NCMS-H and NCMS-L, and the explanations have also been modified accordingly.

Comment 2: In the experimental section, detailed test instruments used for electrochemical tests should be given as well as for material characterization. Furthermore, the way to test the electrochemical properties should be depicted in detail.

Response: The details of the electrochemical test have been added to the “Supporting Information.”  

Comment 3: After being stopped for one week during the cycling process, as shown in Figure 7d, the retention of NCMS-L increased initially while seems to have a tendency to continue declining after 45000 cycles. Therefore, the claim that the capacitance of NSMS-L can hold to 192% after 45000 cycles without prerequisite description is challenging. A longer-term cycle of data should be given to support this opinion, or the pause can be conducted earlier.

Response: The enhanced cycling stability of the composite was because of the electrochemical reconstruction during cycling. The relevant reference has been added to the manuscript (ref. 32). The pause of the electrochemical test re-activated the electrode, as confirmed by the electrochemical cycling test of bare MoS2. A more relevant explanation has also been included in the revised manuscript.

Comment 4: Since the authors set up two products with different Mo contents that generated different morphologies, the electrochemical properties of NCMS-H should be given to compare with NCMS-L in the section results and discussion.

Response: The electrochemical characteristics of NCMS-H were already in the supporting information (Figure S6).

Comment 5: The manuscript should be revised thoroughly because there are several inappropriate details. For example, the ‘NCMS-H’ in line 123, the ‘NoiCo2S4’ in line 331, etc.

Response: Thanks for checking the manuscript. The manuscript has been checked carefully and revised accordingly.

Comment 6: The prepared composites have different energy storage mechanisms in the process of charge and discharge. That will have a certain impact on calculation of specific capacitance or coulomb efficiency, please give more details of the above electrochemical data calculation. Specifically, it is not normal for coulombic efficiency to be greater than 100%. Please give a reasonable explanation.

Response: The electrochemical reactions during the charge/discharge processes have been well supported by the relevant references. The manuscript has already given the equations for calculating the electrochemical data.    

It is well-known that the coulombic efficiency of electrode material is low at high current densities due to the short discharge time and the fast transport of electrolyte ions. On the other hand, at low current densities, the electrodes display coulombic efficiency of >100% because of the slow transport of electrolyte ions, some additional electrode reactions, and the decomposition of electrolytes. Many published works have shown the coulombic efficiency > 100% (refs. 1-3)

1) Zhao, C., Zheng, W., Wang, X., Zhang, H., Cui, X., & Wang, H., “Ultrahigh capacitive performance from both Co(OH)2/graphene electrode and K3Fe(CN)6 electrolyte,” Sci. Rep., 3(1), 1-6 (2013).

2) Liu, L., Zschiesche, H., Antonietti, M., Daffos, B., Tarakina, N. V., Gibilaro, M., ... & Simon, P. “Tuning the Surface Chemistry of MXene to Improve Energy Storage: Example of Nitrification by Salt Melt.” Adv. Energy Mater., 2202709 (2022).

3) Zhai, Z. B., Huang, K. J., & Wu, X., “Superior mixed Co-Cd selenide nanorods for high performance alkaline battery-supercapacitor hybrid energy storage,” Nano Energy, 47, 89-95 (2018). 

Comment 7: XPS analysis was performed in Fig.5 to obtain the valence state of Ni/Co/Mo/S, please provide evidence or relevant reference.

Response: The relevant references are included in the manuscript [20, 23-25] and explained in the main text. 

Comment 8: In line 256 of the text, the authors described the oxidation processes of Ni2+ and Co3+, which were contradictory with the reaction mechanisms of Ni2+ and Co2+ given in the equations (4-6). In addition, XPS analyses show bivalent and trivalent Ni/Co, whether both bivalent and trivalent Ni/Co will react? Is the Faradaic reaction potential of cobalt sulfide consistent with nickel sulfide?

Response: As mentioned in Eq. 4, Co2+ was first converted to Co3+ in its reaction with OH, and then Co3+ was converted to Co4+ during the charging process (Eq. 5). In the case of Ni, Ni2+ was converted to Ni3+ during the charging process (Eq. 6). Therefore, it can be said that the bivalent Co, trivalent Co, and bivalent Ni were involved in the electrochemical reaction. Some previous reports also support such electrochemical reactions (refs. 1-3):

  • Xiao, J., Wan, L., Yang, S., Xiao, F., & Wang, S., “Design hierarchical electrodes with highly conductive NiCo2S4 nanotube arrays grown on carbon fiber paper for high-performance pseudocapacitors,” Nano Lett., 14(2), 831-838 (2014).
  • Jia, Y., Ma, Y., Lin, Y., Tang, J., Shi, W., & He, W., “In-situ growth of hierarchical NiCo2S4/MoS2 nanotube arrays with excellent electrochemical performance,” Electrochim. Acta, 289, 39-46 (2018).
  • Wan, H., Jiang, J., Yu, J., Xu, K., Miao, L., Zhang, L., ... & Ruan, Y., “NiCo2S4 porous nanotubes synthesis via sacrificial templates: High-performance electrode materials of supercapacitors,” CrystEngComm, 15(38), 7649-7651 (2013).

The faradaic reaction potential of cobalt sulfide differs from that of nickel sulfide. Therefore, the CV plot of NiCo2S4 displays different redox peaks.     

Comment 9: Suggest the authors provide pseudo-capacitance calculation and b-value fitting to analyze the capacitance contribution.

Response: I agree with you that discussing the capacitive or pseudo-capacitive contribution is a trend nowadays and also helpful. However, please understand that this study was done 6 years ago. The student who had worked this already returned to his home country a long time back. As his material and results are still attractive, a research professor in my lab recently modified his draft and graphs. It is very difficult for me to get the original data from the former student and ask the research professor to do this kind of calculation after reading the data from the graph.

Comment 10: The authors should also provide the information about the mechanism of hybrid asymmetric supercapacitor.

Response: The hybrid asymmetric supercapacitor displayed a faradaic charge storage mechanism, confirming the presence of redox peaks in CV curves. The explanation has been included in the revised manuscript, too.

Reviewer 3 Report

The manuscript investigates the NiCo2S4/MoS2 composite used for supercapacitors with good coulombic efficiency, high capacitance and energy density, and good cycling stability. This is an interesting work for readers in this field. I would suggest this paper be published after the following concern is addressed.

During the third stage of the composite preparation, the concentration of Mo6+ can significantly affect the morphology of the product. The authors should explain the mechanism behind this in detail.

Author Response

Reviewer #3

Comment: During the third stage of the composite preparation, the concentration of Mo6+ can significantly affect the morphology of the product. The authors should explain the mechanism behind this in detail.

Response: As suggested by the reviewer, the mechanism for the change in morphology has been included in the revised manuscript with a related reference (p. 3).  

Round 2

Reviewer 1 Report

The authors revised satisfactorily.

Reviewer 2 Report

I am satisfied with the author 's reply. The paper can be published now.